# Tracheal branching in ants is area-decreasing, violating a central assumption of network transport models

Ian J. Aitkenhead[1], Grant A. Duffy[1], Citsabehsan Devendran[2], Michael R. Kearney[3], Adrian Neild[2], Steven L. Chown[1]*

**1** School of Biological Sciences, Monash University, Australia, **2** Department of Mechanical and Aerospace Engineering, Monash University, Australia, **3** School of BioSciences, The University of Melbourne, Australia

* steven.chown@monash.edu

**Data Availability Statement:** All data files are available from the Monash Figshare repository (doi: 10.26180/5e449889f25d0).

## Abstract

The structure of tubular transport networks is thought to underlie much of biological regularity, from individuals to ecosystems. A core assumption of transport network models is either area-preserving or area-increasing branching, such that the summed cross-sectional area of all child branches is equal to or greater than the cross-sectional area of their respective parent branch. For insects, the most diverse group of animals, the assumption of area-preserving branching of tracheae is, however, based on measurements of a single individual and an assumption of gas exchange by diffusion. Here we show that ants exhibit neither area-preserving nor area-increasing branching in their abdominal tracheal systems. We find for 20 species of ants that the sum of child tracheal cross-sectional areas is typically less than that of the parent branch (area-decreasing). The radius, rather than the area, of the parent branch is conserved across the sum of child branches. Interpretation of the tracheal system as one optimized for the release of carbon dioxide, while readily catering to oxygen demand, explains the branching pattern. Our results, together with widespread demonstration that gas exchange in insects includes, and is often dominated by, convection, indicate that for generality, network transport models must include consideration of systems with different architectures.

## Author summary

A fundamental assumption of models of the transport of substances through networks of tubes, such as circulatory systems in animals and vascular systems in plants, is that the total cross-sectional area of the tubes remains constant irrespective of the branching level, or that it increases slightly in the direction from the largest to the smallest tubes. One large tube should have the same or a slightly smaller area than the sum of the next two tubes after a branching. The assumption of such a pattern underpins one of biology's most influential ideas–the metabolic theory of ecology. Surprisingly, the assumption has never been systematically examined for insects–the planet's most diverse group of animals which deliver oxygen to and remove carbon dioxide from their bodies using a network of tubes

**Funding:** This research was supported by a grant from the Australian Synchrotron part of the Australian Nuclear Science and Technology Organisation (https://www.ansto.gov.au/research/facilities/australian-synchrotron/overview), and was supported by the Multi-modal Australian ScienceS Imaging and Visualisation Environment (MASSIVE) (www.massive.org.au), through award AS152/IM/9290 to GAD and SLC; and by Australian Research Council (https://www.arc.gov.au/) Discovery Project DP140101240 to MRK and SLC. GAD is the recipient of an Australian Research Council Discovery Early Career Researcher Award (DE190100003). The funders played no role in the study design, data collection and analysis, decision to publish, or preparation of the manuscript. We received technical assistance at the Australian Synchrotron from Chris Hall, Anton Maksimenko and Andrew Stevenson.

**Competing interests:** The authors have declared that no competing interests exist.

known as tracheae. Until recently, it has been technologically very challenging to do so. Here, we use x-ray synchrotron tomography to overcome this challenge. We show that tracheal branching in 20 species of ants does not follow this pattern. Rather, cross-sectional area reduces in an inwards direction. We then use modelling to show that such a pattern facilitates outward $CO_2$ release, a process more challenging for insects than moving oxygen inwards. Our work suggests that much still needs to be done to understand the fundamental assumptions underlying network transport models and how they apply more generally across life–especially in the context of why metabolic rate scales with body size.

## Introduction

Mechanistic explanations for the scaling of energy and mass flows with organism mass have proliferated in the last two decades. The long-term goal has been to find a general mechanism to explain the tendency for quarter-power scaling of metabolic rate as well as departures from such scaling, which would provide a powerful theoretical basis for inferring how energy and mass flow at scales below and above the individual [1–4]. A range of explanations has been proposed for both inter-specific and intra-specific scaling patterns, including quantitative theories incorporating the role of surface-area/volume dynamics of storage pools and the overhead costs of growth, and arguments about the relative roles of supply and demand processes in the context of metabolic activity levels [1,5–9].

The most controversial and high-profile ideas in recent times are transport network-based models [1,7,10–12]. These models make a series of assumptions about the physical characteristics of transport networks (e.g. animal circulatory and plant vascular systems), with the overall assumption that they constrain resource delivery and hence metabolism [10,12]. An advantage of such physically-explicit models is that they make testable predictions about underlying processes and structures [9]. Perhaps for this reason, they have generated substantial debate. A wide range of works has examined empirical evidence for their assumptions and predictions, and the theory underpinning them, noting that these models do not apply in all circumstances, and often conclude that other approaches have greater support [13–18].

The most influential of the transport network-based approaches, the metabolic theory of ecology (MTE) [1,2], assumes that in the branching of plant vessels and insect tracheae (the main transport route for $O_2$ to insect tissues and for $CO_2$ efflux from the body [19]), the total cross-sectional area remains constant across all levels (Da Vinci's rule). In mammals this pattern applies to large vessels, with a transition to an increase in total cross-sectional area of smaller vessels (also known as Murray's law) [1,18]. The summed cross-sectional area of the child branches is either the same as or larger than the cross-sectional area of the parent branch (Fig 1). Support for these assumptions and the consequences of their relaxation have been well explored for mammals and plants [10,12,20–22].

What the situation is for insects remains poorly known. The MTE assumes that in insects tracheal branching is area-preserving and gas exchange is dominated by diffusion [1,23]. Many studies have investigated insect gas exchange and demonstrated convection is common and can dominate gas exchange [24,25]. By contrast, the empirical evidence on which the area-preserving assumption of the MTE is based comes from 100-year old observations of a single caterpillar [26] (S1 Text). Only a few other investigations of the relationship between the radii of parent and child tracheal branches have been undertaken, but they are not widely appreciated. Therefore, the general principle of the way in which network architecture contributes to gas exchange in the Earth's largest group of terrestrial animals remains untested [27].

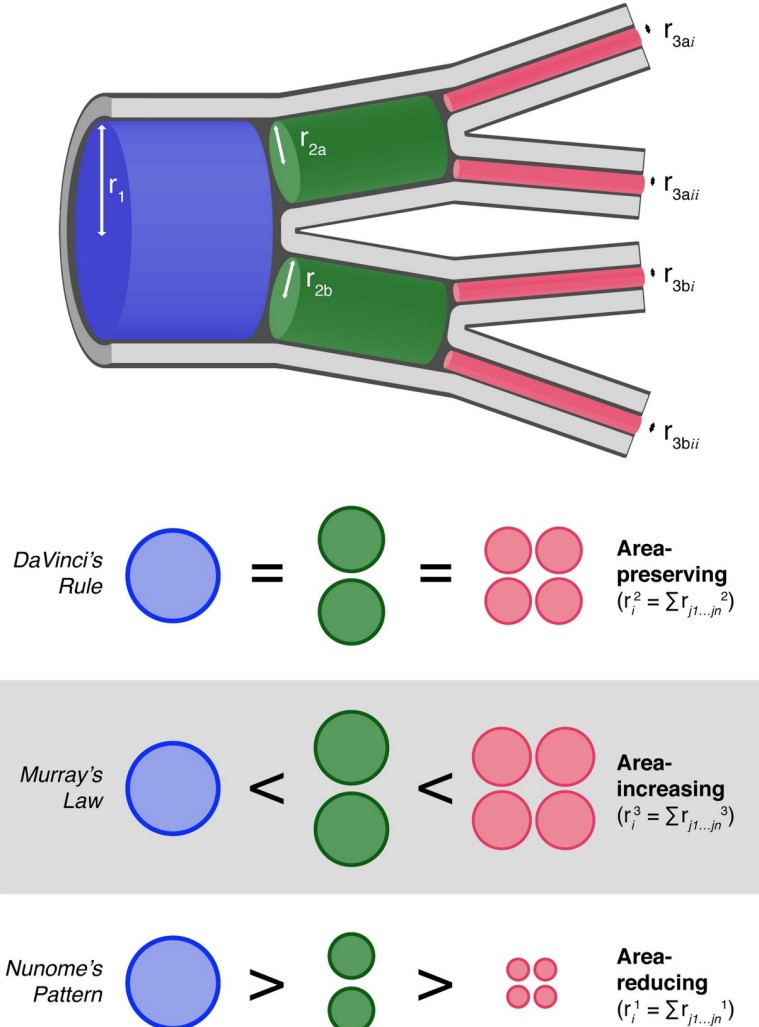

**Fig 1. Diagrammatic representation of transport-network branching following each of the three main branching schematics.** The metabolic theory of ecology assumes that in branching of plant vessels and insect tracheae, cross-sectional area is preserved across branching levels (DaVinci's Rule) [1]. In mammals, large vessels are area-preserving with a transition to area-increasing (Murray's Law) as vessel size decreases [1]. By contrast, Nunome [32] proposed that insect tracheae may be area-reducing (Nunome's Pattern).

We examined this fundamental assumption about insect tracheal network structure by conducting a review of the limited available information and then complementing it with a systematic investigation of cross-sectional area change during abdominal tracheal branching in 165 individuals of 20 species of ants using synchrotron x-ray microtomography [28,29]. Specifically, we tested the prediction that in the equation relating parent to child branches:

$$r_i^\alpha = r_{ja}^\alpha + r_{jb}^\alpha \tag{1}$$

where $r_i$ and $r_j$ are the radii of the parent and child vessels, respectively [21], $\alpha$ should be ~2 for area-preserving branching or ~3 for area-increasing branching, as assumed by the MTE and other theoretical studies [1,23], and acknowledging that variation may be continuous between these values in some circumstances. We also determined whether relationships between parent and child branches changed in concert with changes in radius size of the primary level tracheae

across all of the species examined, by regressing the left- and right-hand sides of Eq 1 against each other. Finally, we modelled $CO_2$ transport in the system, determining what the consequences are of different values of $\alpha$, because tracheal systems are typically confronted by differences in the ease of transport of $CO_2$ relative to transport of $O_2$ [30,31].

## Results

### Tracheal architecture

The ant tracheal system was readily reconstructed from the tomography, revealing the ubiquitous presence of air sacs across the species investigated (Fig 2). Solving for $\alpha$ resulted in a median of $\alpha = 1.01$ for level 1 to 2, and $\alpha = 1.01$ or $1.03$ for level 2 to 3. Data were typically right-skewed (Skewness for Level 1–2: 2.03; L2-3: 0.43; L2-3b: 2.41). After $\log_{10}$ transformation to account for the skew [21], $\alpha$ differed statistically from the expected values of 0.30 (log of 2) or 0.48 (log of 3) (Level 1–2: 0.3: t = -34.7, df = 164, 0.48: t = -58.0, df = 164; L2-3: 0.3: t = -41.2, df = 157; 0.48: t = -65.3, df = 157; L2-3b: 0.3: t = - 25.6, df = 151; 0.48: t = -42.3, df = 151; p < 0.0001 in all cases), but was either identical or close to 0 (95% C.I.: Level 1–2: 0.02 to 0.05; L2-3: -0.02 to 0.01; L2-3b: 0.0 to 0.05), which accords with an untransformed $\alpha = 1$.

The regression analyses revealed no difference with size, and strong linear relationships, though these did not accord with the assumptions of $\alpha = 2$ or $3$, but rather with the expectation that $\alpha = 1$ across the size spectrum [see Materials and Methods] (body length range among individuals: 4–24 mm) (Fig 3A and 3B and S1 Table). We also tested for a species-level effect,

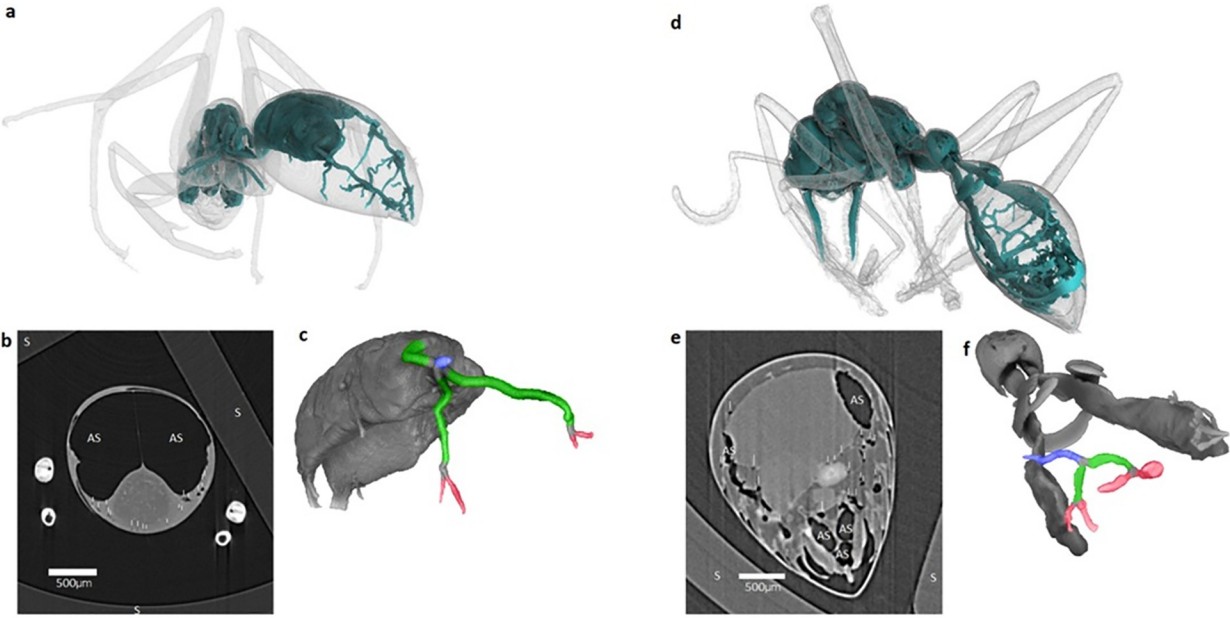

**Fig 2.  Results of the reconstruction of x-ray tomography of a *Camponotus suffusus* minor (a-c), and *Myrmecia fulvipes* (d-f).** (a, d) Volume render of the body and internal airspaces (cyan). (b, e) Horizontal planar slice of the abdomen with air sacs (AS) and tracheal branches (white vertical line) indicated, the mounting syringe (S) is visible around the edges of the images. Additional examples of the raw slices for other ant genera can be found in S3 Fig. (c, f) Volume render of the internal airspace showing the air sac and associated trachea branches, coloured by their respective level as in Fig 1 (blue, green, red for tracheal levels 1, 2, 3, respectively, as in Fig 1). Note perspective distortions arising from presenting a three-dimensional volume in two dimensions. The radii of each branch shown for *C. suffusus* (c) were measured as 29.15 μm (blue), 13.87, 10.11, 5.59 μm (green), and 6.35, 8.02, 5.40, 5.24 μm (red). The radii of each branch shown for *M. fulvipes* (f) were measured as 32.83 μm (blue), 17.67, 14.44 μm (green), and 11.78, 8.13, 6.19, 6.61 μm (red).

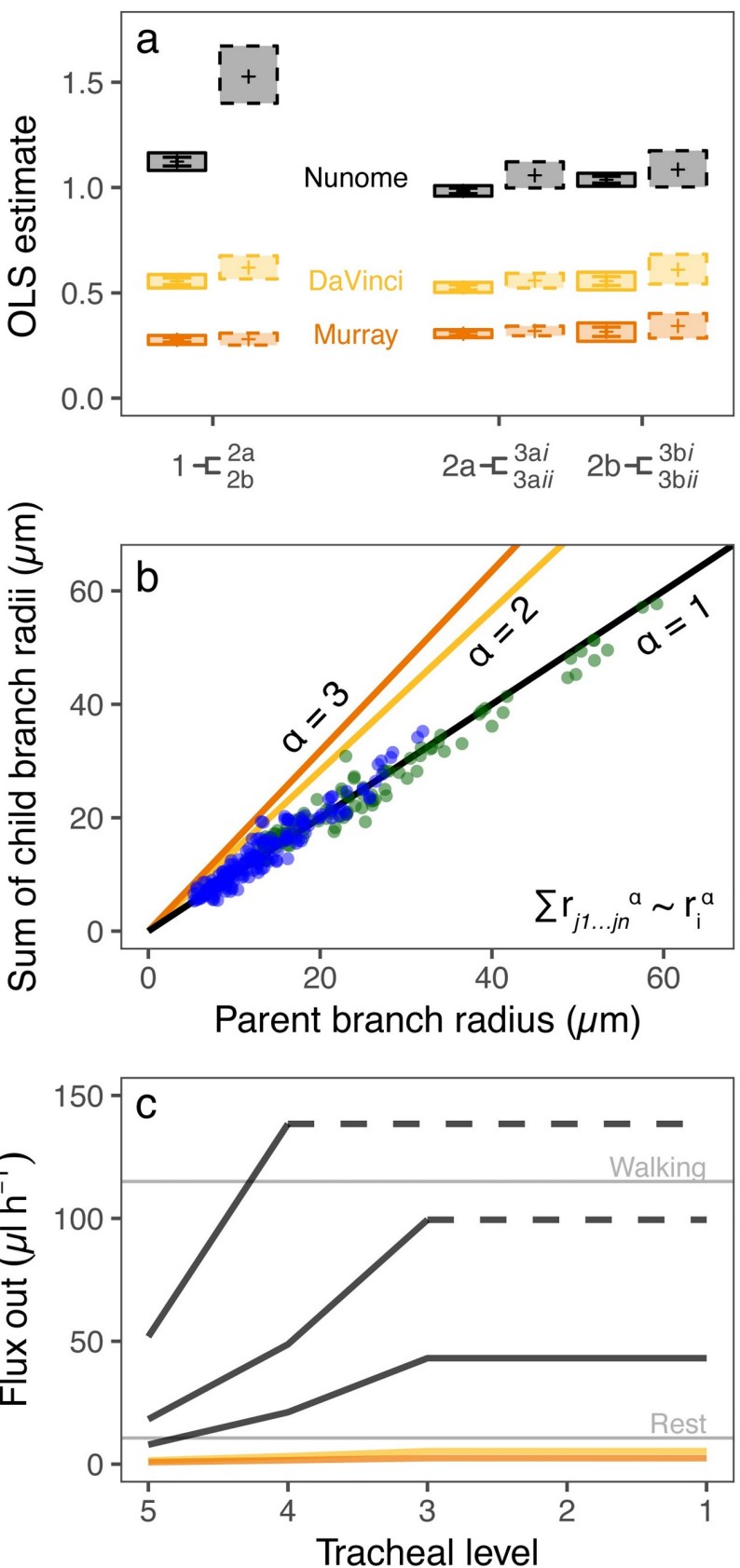

**Fig 3. Predictions and empirical data for the relationship between parent and child branch radii and their modelled consequences for $CO_2$ efflux.** (a) Empirical outcomes (slopes ± standard error) of Ordinary Least Squares (OLS) regression forced through the intercept (sum of radii of child branches$^\alpha$ ~ radius of parent branch$^\alpha$ where $\alpha$ = 1, 2 or 3 for Nunome's, DaVinci's or Murray's laws, respectively; left hand points, solid lines) and Major Axis (Model II) regression (right hand points, dashed lines) for levels 1 to 2, and 2 to 3 (full results in S1 Table). Shaded boxes indicate upper and lower confidence intervals of each model fit and error bars indicate the standard error of the forced intercept models. A slope of approximately 1 is expected if the relationship fits observed data. (b) Relationship between parent and child tracheal branch radii for levels 1–2 (green dots) and 2–3 (blue dots), with the black fitted line indicating an ordinary least squares slope of ~1 ($\alpha$ = 1), whereas the yellow and orange lines indicate expectations for relationships with area-conserving ($\alpha$ = 2) and area-increasing ($\alpha$ = 3) branching, respectively. (c) Modelled whole-organism $CO_2$ flux at each tracheal level (level 5 is tissue level, level 1 is spiracle) for area-conserving (yellow line), area-increasing (orange line), or area-reducing (black lines) transport networks. Each of the three black lines represents a model system with either no air sac (continuous solid line) or an air sac at either the 3$^{rd}$ or 4$^{th}$ level (indicated by the dashed line-segment), after which hydraulic resistance decreases and efflux is projected rather than modelled explicitly. Tracheal branch lengths of the modelled system ranged from 211 μm at the deepest level (level 5) to 1569 μm at level 1 (levels 4, 3, 2 had branch lengths of 352, 586, and 976 μm, respectively). The tracheal radius of level 5 was set at 3 μm. Whole-organism flux was modelled with a partial pressure difference of 6 kPa (see S1 Fig for 4 kPa and 8 kPa) and assuming 6 pairs of abdominal spiracles. Horizontal grey lines indicate the flux needed to meet the metabolic requirements of an average ant at rest or walking (S5 Table). Where flux values are below these activity lines, flux of the system is inferred to be insufficient to maintain activity by diffusion alone.

but found that it was small, resulting in slopes that were generally indistinguishable from those in the OLS regressions (S2 Table).

Therefore, in the ant abdominal tracheal systems examined here, total cross-sectional area is not preserved, nor does it increase, from parent to child branches. Rather, cross-sectional area declines: the summed radii of branches at the lower level equals the radius of the branch at the higher level. We call this 'Nunome's pattern', after early work referring to this form of area change with branching [32].

## $CO_2$ transport modelling

Our modelling (S2 Text) uses tracheal network data from the current investigations and partial pressure variation data from the literature [31,33]. Furthermore, we assume that gas exchange is dominated by diffusion in ants at rest, in accordance with empirical data [34] and the assumptions of the MTE [1]. The modelling outcomes indicate that tracheal architecture with an inwards decline in cross-sectional area with branching (which means area-increasing branching outwards), delivers the highest potential $CO_2$ flux. This $CO_2$ flux is in keeping with empirical data on ant metabolic requirements (Fig 3C and S1 Fig). By contrast, area-preserving branching and Murray's law cannot deliver the required $CO_2$ efflux.

A model of convection presuming that convection is periodic, as is true of many insects [24], indicates that an additional 0.241 μl h$^{-1}$ of $CO_2$ flux per spiracle can be achieved when the network is characterised by inwards area-reducing branching, under the assumption of 66% volume evacuation at a frequency of 0.26 Hz (tracheal compression parameters based on empirical observations [35]). This value contributes to the background flux provided by the diffusive system (Fig 3C) and, as flux from this model scales linearly with evacuation frequency, flux gains through convection are expected to increase as needed as individuals become more active.

## Discussion

Our findings contrast strongly with those of Krogh [26], the foundation on which contemporary network models of metabolic ecology are based [1], that cross-sectional area is conserved (S1 Text). They are, however, in keeping with overlooked studies on an adult dragonfly [36], in part also with data on a water bug and silkworm [32,37], and with recent data for tracheae

of *Drosophila melanogaster* [38]. For the 30 species of insects that have been examined for cross-sectional area variation among tracheal branch levels, in 22 of them total cross-sectional area of measured tracheae declines with branching, in six cross-sectional area is preserved, and in two species the pattern is variable (S3 Table).

Current theory dictates that either area-preserving or area-increasing branching results in the required resource delivery by tubular transport networks [1,12,18]. Yet, the ants examined here and some other insects have evolved an entirely different branching pattern in their tracheal system. In consequence, such a transport network should provide a physiological advantage proximally, and ultimately a fitness benefit. What that advantage might be is not immediately obvious given long-standing statements that any system capable of delivering an oxygen supply to the tissues should be adequate for the removal of $CO_2$ [19]. Those statements are typically predicated, however, on the basis of tissue solubility differences between the two gasses, rather than also on the way they are transported to and from metabolically active areas through the tracheal system. Oxygen is generally delivered from the spiracles along the tracheae to the finest tracheole branches which lie within a few micrometres of the mitochondria where it is required [19,31]. By contrast, $CO_2$ accumulates within the insect and is buffered in the haemolymph. It then moves across the tracheal walls back into the tracheal lumina across the full length of the tracheal system, rather than just at the finest tracheole branches [30,39]. As a consequence of this situation, and partial pressure differences between intratracheal and external $O_2$ and $CO_2$, respectively, tracheal systems are confronted by too little transport of $CO_2$ relative to transport of $O_2$ [30,33], and are sensitive to much smaller experimental changes in the partial pressure of $CO_2$ than of $O_2$ [31]. In consequence, tracheal systems are likely to be optimised for $CO_2$ transport outwards.

Our model of diffusion (S2 Text) reveals why optimisation for outward $CO_2$ transport is likely the case. Nunome's pattern delivers the highest potential $CO_2$ flux: enough to meet the metabolic demands of individual ants. Area-preserving branching and Murray's law either are less adequate, or entirely unable to do so, respectively. Thus, they seem less likely to be found in insects, especially in the case of Murray's law, because such branching patterns cannot deliver the required $CO_2$ efflux, accounting for the empirical data demonstrating that Murray's law is rare in insects (S3 Table).

The ant tracheal system includes air sacs, as both our investigations (Fig 2) and others [27] have revealed. They are likely to be involved in some form of convective gas exchange as has been demonstrated for many other insects [24,25], but are not explicitly included in the network geometry considered by the MTE. Our model of convection indicates that considerable additional $CO_2$ flux per spiracle can be achieved when the network is characterised by Nunome's pattern, with further potential increases associated with increasing organismal activity. Where external $CO_2$ concentrations are high, such as in underground nests or other locations with $CO_2$ accumulation [40,41], Nunome's pattern may be the only architecture that enables sufficient gas exchange (Fig 3C). Under the most extreme circumstances, however, ants are compelled to switch to anaerobic metabolism [42].

Although we recognise that gas flux in the tracheoles might have greater complexity than we have indicated here [43], our modelling and empirical data provide an initial demonstration of why tracheal network architecture in ants follows Nunome's pattern. Our data for the more proximal tracheal branches also show that such branching patterns are not simply restricted to more distal elements of the system as suggested recently [38]. They demonstrate clearly why, in insects, tracheal architecture is unlikely to conform to Murray's law. That is, the system has evolved to meet the demands of both $O_2$ influx and $CO_2$ efflux. The ca. 133 million-year separation between the ants measured here and their last known common ancestor [44] (S2 Fig) suggests that branching in accordance with Nunome's pattern may be ubiquitous

across all ant species. Further empirical investigations, which are now within relatively easy reach given the availability of x-ray microtomography [28,29], coupled with additional modelling, will reveal whether Nunome's pattern is most typical for insects which encounter high $CO_2$ environments, while Da Vinci's rule or some more complex arrangement [38] dominates in others. Irrespective, two major empirical assumptions of the metabolic theory of ecology, that branching is either area-preserving or area-increasing, and that gas exchange is predominantly by diffusion, do not apply routinely to insects, the most diverse group of animals globally.

What are the implications of our findings for the mechanistic role of branching networks in theories of metabolic scaling? It has been raised previously that the generality of the mechanistic role of branching networks in such theories is limited by the absence of closed circulatory systems in many groups [45]. Until the implications of varying architecture of branching systems, like that described here, are explored further, the generality of network transport-based models will remain further reduced. Explorations for some aspects of architectural variation have already begun [10–12]. The findings presented here indicate that broadening this exploration to encompass area-reducing architecture is an intuitive next step as network-based transport models, and the MTE in particular, continue to explain a wide range of ecological phenomena [46,47]. Examining tracheal architecture in ants in concert with metabolic rate scaling and cell size variation would likely prove especially beneficial given previous outcomes from investigations of the latter [14].

## Materials and methods

We used synchrotron x-ray microtomography [28,29] to measure the cross-sectional area of the abdominal tracheae of 165 individuals from 20 species of ants. We use ants as the focal taxon because: (i) the characteristics of gas exchange and metabolic scaling have been relatively well characterized for them, including demonstration of a predominance of diffusion at rest, (ii) substantial size variation is found both within and among species, and (iii) because they are an ecologically important group of insects [14,34,48,49].

### Collection and initial treatment

Individual ants of twenty species in the genera *Camponotus* (8 spp.), *Rhytidoponera* (6 spp.) *Myrmecia* (4 spp.), *Polyrachis* (1 sp.) and *Iridomyrmex* (1 sp.) were collected from sites in Victoria and Queensland, Australia. Samples from Victoria were kept alive in 100 ml vials on moistened plaster of Paris substrate until their return to the laboratory. Ants were kept in these vials for up to four days at 20˚C in a Panasonic MLR-353H-PE climate chamber (Panasonic Healthcare Co., Ltd, Sakata) with a 12:12 L:D cycle. They were then exposed to iodine vapour for 7 days to improve contrast for imaging [50], which resulted in death. Ants from Queensland were exposed to iodine vapour within 8 hours of collection such that preserved specimens could be returned to our laboratory without compromising inter-state biosecurity protocols relating to transport of live insects. Species were identified *post mortem* using relevant keys [51,52], with subsequent specialist verification (S4 Table). Up to 12 ants were loaded into customised 10 ml plastic syringes in preparation for synchrotron x-ray tomography at the Australian Synchrotron.

### Tracheal imaging and measurements

Synchrotron x-ray image data were collected on hutch-B of the imaging and medical beamline (IMBL) at the Australian Synchrotron. The IMBL custom-designed 'Ruby' detector, comprising a PCO.edge sensor equipped with a Nikon Micro-Nikkor 105 mm/f 2.8 macro lens and a

scintillator with a 200 μm terbium-doped gadolinium oxy-sulphide screen [53], was used for imaging. The distance between sample and detector was 50 cm. Each syringe containing ants was mounted between the source and detector and scanned at an energy of 35 KeV. X-ray projections were taken every 0.1˚ over an 180˚ rotation. Individuals were scanned a single time only, with scans lasting from 3–6 minutes per individual. Resolution of the images (7.6 μm) was determined by measuring an object of known size (16.05 mm), measured with Mitutoyo-Absolute Series 500 digital callipers (instrument error ± 0.02 mm; Mitutoyo, Kawasagi).

Raw tomography data were reconstructed using the XLI-CT workflow V0.9.6 software (CSIRO, Clayton, https://www.ts-imaging.net/), where vertical stacking was required the overlap was 110 pixels; no measurements were made in areas of overlap. Individual ants were then manually segmented from resultant image stacks using the freehand cropping tool in Fiji image analysis software [54]. Individual stacks were then binarized using the triangle method [55] and the binarized image was inverted to allow for modelling of the airspace rather than the animal tissue. Internal airspaces were then mapped by selecting a single point in the abdominal tracheal system and using the *Find Connected Regions* Fiji plugin. This process produced a second binary image stack of only the internal airspaces. This internal-only image stack was imported into Avizo 9.0.0 (FEI Visualization Sciences Group, Oregon, http://www.vsg3d.com/) and a volume render of the image stack was created to enable measurement of the airspace in three-dimensional space (Fig 2).

Measurements were made on a single abdominal branch on the left-hand side of each animal (Fig 2). The diameter of each branch was measured as the largest distance between the internal tracheal walls of the primary, secondary and tertiary branches (Figs 1 and 2) using Avizo's 3D length measuring tool. Values were multiplied by a scaling factor of 7.6 to convert them from distance in pixels to μm. Bifurcations (100/165 measured branches) and less commonly three (61/165) and four (4/165) branches were found in the tracheal architecture. Branches with three or more sub-branches were found exclusively in the genus *Camponotus*. Given previous assessments that some asymmetry does not alter assumptions of area-preserving or area-increasing branching [12] we included measurements of all branch-types in our analyses. All tracheal diameter measurements were made manually by the first author (IA) and then halved to represent radii. Measurements were validated by remeasuring the full dataset and comparing first and second measurement values: across the full data set of 1258 measurements, no measures differed statistically by more than 0.15 μm (below the nominal resolution of the system of 7.6 μm), with a mean difference of 0.05 μm. The original measurements were used for analysis.

## Statistical analysis

The assumptions of area-preserving or area-increasing tracheal network branching were examined in two ways. First, we estimated the exponent ($\alpha$) in the equation:

$$r_i^\alpha = r_{ja}^\alpha + r_{jb}^\alpha \tag{2}$$

where $r_i$ is the radius of the branch at parent level $i$ and $r_j$ the radius of the bifurcating branches $a$ and $b$ at subsidiary level $j$ [21], using the *uniroot* function in R (v3.5.0) [56], with an expansion to include additional branches in the case of more complex branching patterns (e.g. trifurcations). In some insects, the relationship between features among levels and in different parts of the tracheal system can change [36,37]. We therefore used data for all individuals from all species to separately estimate $\alpha$ across step-wise levels, from level 1 to level 2, and then from level 2 to level 3. In the latter case, we examined both sets of branches (i.e. branches a and b on Fig 1) to ensure the pattern of branching is consistent. For illustration we also solved Eq 2 for

all bifurcating branches of individuals across all levels simultaneously. The resulting distribution of $\alpha$ in each case was right-skewed, thus we transformed $\alpha$ ($\log_{10}$) so that empirical values could be reliably tested against expected values (0.30 and 0.48 for $\alpha$ of 2 or 3, respectively) using a t-test in R.

Next, we determined whether relationships between parent (as X) and the summed area of child branches (as Y) changed in concert with changes in radius size of the primary level tracheae across all of the species examined. Using empirical data, the slope of a relationship between the left-hand and right-hand terms of Eq 2 closest to 1 (assuming the intercept passes through the origin), under either area-decreasing ($\alpha = 1$), area-preserving ($\alpha = 2$), or area-increasing branching ($\alpha = 3$), will indicate the branching pattern with the greatest empirical support. To examine whether this was the case, we used model I regression (Ordinary Least Squares), with the intercept forced through the origin, implemented in R, and using Major Axis (Model II) regression because of equal measurement error expected in both the dependent and independent variables [57], using the lmodel2 library in R. In each case we excluded outliers that were clearly outside the size range of most of our data (level 1 to 2: 6 individuals; level 2a to 3a: 9 individuals; level 2b to 3b: 13 individuals). Outlier removal and missing data meant that a total of 165, 159, and 153 individuals were analysed in each case. Although comparison among species was not our primary focus, species level effects were nevertheless tested for by including species as a random term in a generalised linear model implemented in the nlme package of R.

## $CO_2$ transport modelling

A simple model of steady-state diffusive gas exchange was used to elucidate an optimised design for effective outward transport of $CO_2$. The diffusion model was based on the modified version of Fick's law of diffusion as detailed in Kestler [58]:

$$J = \frac{A}{L} K \Delta pp_{,CO_2} \tag{3}$$

where, J = volumetric flux of $CO_2$ (in $m^3$ $s^{-1}$), A = tracheal cross-sectional area (in $m^2$), L = branch length (in m), K = Krogh's constant (assumed to be $1.428104 \times 10^{-10}$ $m^2$ $s^{-1}$ $Pa^{-1}$) [58] and $\Delta pp_{,CO_2}$ is the partial pressure difference of $CO_2$ (in Pa). To obtain the volumetric flux of $CO_2$ in $\mu l$ $h^{-1}$, we multiplied J by a conversion factor $3.6 \times 10^{12}$ (considering; 3600 s per hour and $1 \times 10^9$ $\mu l$ per $m^3$). We treated each tracheal branch level as a fluidic channel in a serial configuration. As two child branches converge and feed into the parent branch upstream (i.e. from Level 5 to Level 1), they influence $CO_2$ transport, akin to hydraulic resistance impeding fluid flow in a typical fluidic system. Accommodating for $CO_2$ influx into the tracheal network from the tissue and integrating along the length, we derived an analytical expression for the complete system (for each spiracle inwards, making the simplifying assumption that there's little connectedness among spiracles), consisting of 5 bifurcating tracheal branch levels.

To derive (full derivation detailed in S2 Text) the diffusive gas transport, we first assumed the radius of the deepest branch (i.e. $r_5$) and the corresponding branch lengths. A model ant with tracheal branch lengths of $L_1$: 1569 $\mu m$, $L_2$: 976 $\mu m$, $L_3$: 586 $\mu m$, $L_4$: 352 $\mu m$, $L_5$: 211 $\mu m$ (total tracheal length of 3694 $\mu m$) and tracheal radius at the deepest level (i.e. $r_5$) 3 $\mu m$ was then assumed. Level 5 tracheole radii of this size are in keeping with the available data on insects [36,39]. Second, we assumed that the effective diffusive flux per unit area, $\varphi$ from the tissue into the tracheal branch which is known to occur at level 3–5 [39] is a constant of arbitrary value. Finally, we assumed a $CO_2$ partial pressure difference between the deepest branch level (i.e. level 5) and the external environment to be 6 kPa [33,49]. Based on the assumptions

made and using Eq 3, the effective outward flux of $CO_2$ could be evaluated (S2 Text). Assuming the corresponding branch lengths are held constant, we investigated the effective outward volumetric flux for α as 1 (area reducing), 2 (area conserving) and 3 (area increasing) respectively (Fig 3). The influence on net outward flux due to the presence of air sacs, which are distributed across ant tracheal systems (Fig 2) [27], was also modelled. An air sac, which acts to expel the entire air cavity thereafter (i.e. pressure release) was included at the end of the tracheal branch at either level 4 or level 3 of the modelled system. Values were summed across six pairs of abdominal spiracles in alignment with our observations of number of spiracles in the ants examined. Diffusion-based models for high $CO_2$ environments (i.e. lower $\Delta pp,_{CO2}$ of 4 kPa) and high activity (i.e. higher $\Delta pp,_{CO2}$ of 8 kPa), with all other parameters unchanged, were also created to examine the effects of an altered $CO_2$ diffusion gradient on achievable flux.

In addition, a simplified convection model was used to understand the convective contribution of $CO_2$ efflux when the ants undergo periodic breathing. The model assumed a 66% evacuation of the entire tracheal airway with every exhalation. The frequency of tracheal contraction was assumed to be 0.26 Hz [35], whilst the assumptions used in the diffusive model were retained (i.e. the branch lengths, partial pressure difference and $r_5$ assumed to be the same). The local partial pressures throughout the airway were evaluated as per the diffusive steady-state case (S2 Text) to accommodate for the resultant varying local concentrations of $CO_2$ in the tracheal network.

To compare with modelled estimates of $CO_2$ flux, empirical data on ant metabolic rates were used to estimate $CO_2$ flux (S5 Table). Thirty-eight estimates (from 21 species) of metabolic rate (based on measures of $CO_2$ production or $O_2$ consumption) under a range of temperatures (20–43˚C) for ants undertaking different activities (resting, walking or running workers) were used. Values were converted to $CO_2$ flux rate following Lighton [59]. Based on these data mean empirical $CO_2$ flux rates of: (1) Resting 10.63 μl.h$^{-1}$, (2) Walking/running 115 μl.h$^{-1}$, were used, acknowledging that thermal variation in rates has been included implicitly by using measures made at different temperatures. Tissue-level $CO_2$ flux rate was calculated as $CO_2$ flux rate divided by 70% of mass (assuming that 30% of tissue is metabolically relatively inactive–such as the chitin exoskeleton).

## Supporting information

**S1 Text. Translation of pages 98–100 of Krogh, A. 1920.** Studien über Tracheenrespiration II[1]). Über Gasdiffusion in den Tracheen. Pflügers Archiv 179: 95–112.
(PDF)

**S2 Text. Analytical Derivation of Diffusion.**
(PDF)

**S1 Table. Outcomes of the Ordinary Least Squares (OLS) regression forced through the intercept and Major Axis (Model II) regression for levels 1 to 2, and 2 to 3.**
(PDF)

**S2 Table. Linear mixed-effects model fit by REML—Species as random effect—Forced intercept.**
(PDF)

**S3 Table. Previous measurements of the relationship between parent and child branches in insect tracheae.**
(PDF)

**S4 Table. Ant collection localities.**
(PDF)

**S5 Table. Ant metabolic rate data from the literature.**
(PDF)

**S1 Fig. Effect of altered $pp$CO$_2$ gradient on modelled whole-organism CO$_2$ flux.**
(PDF)

**S2 Fig. Time-calibrated phylogenetic tree for all ant species examined.**
(PDF)

**S3 Fig. Raw, annotated synchrotron x-ray tomography slices from an individual of each genus investigated.**
(PDF)

## Acknowledgments

We thank members of the ChownLab for assistance during the scanning. Jon Harrison and Roger Seymour provided insightful comments on a previous version of the manuscript. Doug Glazier and Jake Socha provided valuable comments during the review process.

## Author Contributions

**Conceptualization:** Ian J. Aitkenhead, Grant A. Duffy, Steven L. Chown.

**Data curation:** Ian J. Aitkenhead, Grant A. Duffy, Steven L. Chown.

**Formal analysis:** Ian J. Aitkenhead, Grant A. Duffy, Citsabehsan Devendran, Michael R. Kearney, Adrian Neild, Steven L. Chown.

**Funding acquisition:** Grant A. Duffy, Michael R. Kearney, Steven L. Chown.

**Investigation:** Ian J. Aitkenhead, Grant A. Duffy, Citsabehsan Devendran, Michael R. Kearney, Adrian Neild, Steven L. Chown.

**Methodology:** Ian J. Aitkenhead, Grant A. Duffy, Citsabehsan Devendran, Adrian Neild, Steven L. Chown.

**Writing – original draft:** Grant A. Duffy, Steven L. Chown.

**Writing – review & editing:** Ian J. Aitkenhead, Grant A. Duffy, Citsabehsan Devendran, Michael R. Kearney, Adrian Neild.

**Writing – review – editing:** Steven L. Chown.

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
