## [Decision Letter · Decision Letter 0]

16 Dec 2019

Dear Dr Chown,

Thank you very much for submitting your manuscript 'Tracheal branching in ants is area-decreasing, violating a central assumption of network transport models' for review by PLOS Computational Biology. Your manuscript has been fully evaluated by the PLOS Computational Biology editorial team and in this case also by independent peer reviewers. The reviewers appreciated the attention to an important problem, but raised some substantial concerns about the manuscript as it currently stands. While your manuscript cannot be accepted in its present form, we are willing to consider a revised version in which the issues raised by the reviewers have been adequately addressed. We cannot, of course, promise publication at that time.

Sincerely,

Van M Savage

Guest Editor

PLOS Computational Biology

Stefano Allesina

Deputy Editor

PLOS Computational Biology

[LINK]

The reviewers both appreciate the contributions of this paper and recognize the incredible value of the empirical data and the amount of work it took to obtain these measurements. The connection to allometric theory is also viewed as important. Indeed, the common theme among the reviewers is that there should be more discussion of the details and debates about metabolic theory--though for somewhat different reasons and points. I think the general advice is good. Test acknowledging limitations of allometric theory, that debates exists, the variation exists and possible sources of this variation, and how this all connects to the data in the present paper would be welcome. However, I think this can be kept fairly brief via modifications to existing text and perhaps a few additional paragraphs throughout. I don't think the main themes of the paper needs to change, and I think leading with the main results and dominant views of metabolic theory is appropriate. All other comments by the referees should be addressed and changes made to the paper where appropriate, including correction of typographical errors.

Reviewer's Responses to Questions

**Comments to the Authors:**

Reviewer #1: General comments:

The authors show that the tracheal transport networks in 20 species of ants are area decreasing, rather than area preserving or slightly increasing, as assumed by resource-transport-network models at the core of the influential metabolic theory of ecology (MTE). Technically, this is a remarkable piece of work. The authors’ findings and their implications for O2 uptake and CO2 release are both useful and noteworthy because they apply to the most diverse major taxon of life, the insects.

However, I have two major concerns, which I believe that the authors should consider.

1) It would be helpful if the authors explicitly addressed the question of whether and how the geometry and physics of transport networks affect the body-mass scaling of metabolic rate. As the authors know, this is a controversial topic, but in my opinion, they do not give sufficient context so that readers can fully appreciate the many dimensions of this controversy and the relevance of their own findings. I am not suggesting a long discussion about this. However, it would be helpful if readers were made aware that:

(a) Closed resource-transport networks, assumed by the MTE, do not occur in all organisms. Many groups of animals and plants do not have closed resource-transport networks, and some do not have any anatomical transport networks at all. Therefore the MTE, as originally proposed, has limited applicability.

(b) Insects have air sacs in their tracheal transport networks, which departs from the network geometry specified by resource-transport-network models at the core of the MTE.

(c) Resource-transport-network models of metabolic scaling currently have no direct mechanistic empirical support, unlike some other models based on surface-area exchange of resources/wastes, and size-related changes in cell size/number, various resource-demanding processes, and the relative amounts of tissues with varying metabolic intensities (reviewed in Glazier 2014, 2018a).

(d) Several lines of evidence contradict predictions of resource-transport-network models of metabolic scaling, or are not explained by these models (see e.g., Chown et al. 2007; Glazier 2014, 2015, 2018a; Hirst et al. 2014; Glazier et al. 2015; Harrison 2017).

2) It would be helpful if the authors considered whether the area-decreasing branching geometry of tracheal transport networks in ants helps to explain their metabolic scaling, if at all. In particular, how do the authors’ findings help explain the diverse metabolic scaling relationships shown by various ant species, as reported previously by one of the authors and his colleagues (Chown et al. 2007). Can this variation in metabolic scaling (slopes ranging between 0.56 and 1.28) be explained by parallel variation in the geometry of their transport networks? If not, the claim that resource-transport-network models of metabolic scaling have no direct mechanistic empirical support still stands. No mention is made of the claim by Chown et al. (2007) that the cell-size model better explains the metabolic scaling of ants than do resource-transport-network models.

In the last sentence of their summary the authors state: “Our work suggests that much still needs to be done to understand the fundamental assumptions underlying network transport models and how they apply more generally across life – especially in the context of virtually ubiquitous metabolic scaling.” However, nowhere in the main text do the authors discuss the relevance of their findings for our understanding of metabolic scaling in an explicit way.

Specific comments:

Author summary, L 4, 12: Please clarify what is meant by “inwards direction”. Do you mean from the external spiracles towards the inner body core?

Author summary, L 14-15: What is meant by “virtually ubiquitous metabolic scaling”, Are you referring to the 3/4-power law, or merely to the fact that metabolic rate scales with body mass regardless of the slope. Metabolic scaling may be log-log linear or curvilinear with slopes varying from near 0 to > 1. Whatever way the authors view this, how do they think that the geometry of transport networks affects metabolic scaling relationships, if at all? Evidence is growing that transport networks may not exert supply limits on resting metabolic rate, as assumed by the MTE (see e.g., Glazier 2015; Harrison 2017). Some investigators are now arguing that effects of resource demand should be considered in addition to or instead of resource-supply limits (e.g., Glazier 2014, 2018b; Harrison 2017). The question boils down to whether the geometry of transport networks are primary causes or secondary responses to metabolic scaling.

Methods (p 12): In OLS regression analyses, it is assumed that there is a dependent (Y) and independent variable (X). Is ri or rj the X variable, and if so what is the justification?

Fig. 2b, e: Please make sure that the features designated in these pictures are more easily visible to readers.

Fig. 3a: Are the OLS “outcomes” slopes? If so, please say so. Also please make clear what the X and Y variables are for these regressions. From reading the text (e.g. on P 4), I assume that the variables are ri and rj (radii of the parent and child vessels), but I am not sure which was considered X versus Y. I am having difficulty matching up the data shown in panel ‘a’ with those shown in panel ‘b’.

References:

Chown, S. L., Marais, E., Terblanche, J. S., Klok, C. J., Lighton, J. R. B., & Blackburn, T. M. (2007). Scaling of insect metabolic rate is inconsistent with the nutrient supply network model. Functional Ecology, 21(2), 282-290.

Glazier, D. (2015). Body-mass scaling of metabolic rate: what are the relative roles of cellular versus systemic effects? Biology, 4(1), 187-199.

Glazier, D. (2018a). Rediscovering and reviving old observations and explanations of metabolic scaling in living systems. Systems, 6(1), 4.

Glazier, D. S. (2018b). Resource supply and demand both affect metabolic scaling: A response to Harrison. Trends in Ecology & Evolution, 33(4), 237-238.

Glazier, D. S., Hirst, A. G., & Atkinson, D. (2015). Shape shifting predicts ontogenetic changes in metabolic scaling in diverse aquatic invertebrates. Proceedings of the Royal Society B: Biological Sciences, 282(1802), 20142302.

Harrison, J. F. (2017). Do performance–safety tradeoffs cause hypometric metabolic scaling in animals?. Trends in Ecology & Evolution, 32(9), 653-664.

Hirst, A. G., Glazier, D. S., & Atkinson, D. (2014). Body shape shifting during growth permits tests that distinguish between competing geometric theories of metabolic scaling. Ecology Letters, 17(10), 1274-1281.

Reviewer #2: see attachment

**Have all data underlying the figures and results presented in the manuscript been provided?**

Reviewer #1: Yes

Reviewer #2: Yes

PLOS authors have the option to publish the peer review history of their article (what does this mean?). If published, this will include your full peer review and any attached files.

Reviewer #1: Yes: Douglas S. Glazier

Reviewer #2: Yes: Jake Socha

---

## [Decision Letter · Decision Letter 1]

31 Mar 2020

Dear Dr. Chown,

Thank you very much for submitting your manuscript "Tracheal branching in ants is area-decreasing, violating a central assumption of network transport models" for consideration at PLOS Computational Biology. As with all papers reviewed by the journal, your manuscript was reviewed by members of the editorial board and by several independent reviewers. The reviewers appreciated the attention to an important topic. Based on the reviews, we are likely to accept this manuscript for publication, providing that you modify the manuscript according to the review recommendations.

Reviewer 1 had no additional comments, and Reviewer 2 had a few final comments that the authors should consider in finalizing their manuscript. Both reviewers were very pleased with the revision and with the content and quality of the paper overall.

Sincerely,

Van M Savage

Guest Editor

PLOS Computational Biology

Stefano Allesina

Deputy Editor

PLOS Computational Biology

[LINK]

Reviewer's Responses to Questions

**Comments to the Authors:**

Reviewer #1: Great piece of work! The authors have effectively dealt with all of my concerns. I look forward to the day when direct comparisons between the geometry of resource-transport networks and the scaling of metabolic rate are made across individual species, a huge but necessary task for evaluating the validity of resource-transport network models of metabolic scaling.

Reviewer #2: I apologize to the authors for my delay. The Covid-19 crisis hit the US in the past few weeks, and as with many of us, I scrambled to rework my classes and deal with other things as a consequence.

The authors have made a great number of changes to improve the manuscript, and I have no further major comments. I do have a few minor suggestions below. I stand by my original assessment that this manuscript is an exciting piece of work, and I applaud the authors for their efforts in going above and beyond to provide a great amount of detail, useful scholarship for the field. Congratulations.

Page 2

>“Interpretation of the tracheal system as one optimized for the release of carbon dioxide, while readily catering to oxygen demand, explains the branching pattern.”

In the abstract here, having not read the study, it’s not clear if mechanical optimization or evolutionary optimization is intended. As these mean different things and are analyzed differently, it would be helpful to indicate that here. (I assume mechanical is meant.)

>“Our results, together with widespread demonstration that gas exchange in insects includes, and is often dominated by, convection, indicate that for generality, network transport models must include consideration of systems with bidirectional flow.”

The directionality doesn’t matter, per se. The same argument could apply for grasshoppers, which partially use unidirectional flow.

Page 4

>“models do not apply in all circumstances, and often concluding that other approaches have”

Change ‘concluding’ to ‘conclude’.

Page 8

>”In consequence, tracheal systems are likely to be optimised for CO2 transport outwards.”

Same comment about mechanical optimization applies here.

Page 9

>”Until the implications … is explored further,”

Change to ‘are explored’.

Page 11

Thank you for supplying details of the imaging and providing sample slices in the supplement; it is all useful to see. One more question: what is the nominal resolution of the system? The value of 0.15 µm difference described below should be at or below the possible resolution.

Also, as there are very few studies of tracheal systems using synchrotron tomography, I think it would be useful to point to the first one (Socha and De Carlo 2008), because the methods of preparation described here are different (and were also used in Harrison et al 2018). Of course no strict need to do this; I don’t mean to insist that you cite my paper, by any means.

Page 14

>”A simple model of steady state diffusive gas exchange”

Change to ‘steady-state’.

Supplement 1

I apologize for my original comments about editorial errors in the document. Somehow when I was reading it I missed the header, indicating that it was a translation. Clearly I was reading too quickly, and I thought that the text described a comparison to his original study. I actually think it’s a great service to the field to provide the translation in this paper, so I thank you. (Many years ago, I believe that I sat down with a German speaker to discuss the paper, but I didn’t have a written translation to work with.) For completeness, it would be nice to provide the name of the person who translated it as credit at the top, as well as the full original citation and translated title.

Supplement 6

Just a side note in response to the comment, “We note that Krogh did not supply units, and that ratios have no units.” Ratios are not necessarily dimensionless; for example, speed is a ratio that has units of length per time.

**Have all data underlying the figures and results presented in the manuscript been provided?**

Reviewer #1: Yes

Reviewer #2: Yes

PLOS authors have the option to publish the peer review history of their article (what does this mean?). If published, this will include your full peer review and any attached files.

Reviewer #1: Yes: Douglas S. Glazier

Reviewer #2: Yes: Jake Socha
---

## [Editor Report · Decision Letter 2]

6 Apr 2020

Dear Dr. Chown,

We are pleased to inform you that your manuscript 'Tracheal branching in ants is area-decreasing, violating a central assumption of network transport models' has been provisionally accepted for publication in PLOS Computational Biology.

Best regards,

Van M Savage

Guest Editor

PLOS Computational Biology

Stefano Allesina

Deputy Editor

PLOS Computational Biology

---

## [Editor Report · Acceptance letter]

15 Apr 2020

PCOMPBIOL-D-19-01679R2 

Tracheal branching in ants is area-decreasing, violating a central assumption of network transport models

Dear Dr Chown,

I am pleased to inform you that your manuscript has been formally accepted for publication in PLOS Computational Biology. Your manuscript is now with our production department and you will be notified of the publication date in due course.

With kind regards,

Laura Mallard
